# Discharge against medical advice in Special Care Newborn Unit in Chattogram, Bangladesh: Prevalence, causes and predictors

**Syeda Humaida Hasan** [1]*, **Jagadish Chandra Das**[1], **Kamrun Nahar**[1], **Muhammad Jabed Bin Amin Chowdhury**[1], **Tamanna Zahur**[2], **Mohammad Abu Faisal**[3¤], **Zabeen Choudhury**[4], **Dhiman Chowdhury**[4]

1 Department of Neonatology, Chittagong Medical College, Chattogram, Bangladesh, 2 Department of Public Health, Chittagong Medical College, Chattogram, Bangladesh, 3 Department of Gastroenterology, Cox's Bazar Medical College, Cox's Bazar, Bangladesh, 4 Department of Pediatrics, Chittagong Medical College, Chattogram, Bangladesh

¤ Current address: Department of Gastroenterology, Chittagong Medical College, Chattogram, Bangladesh
* humaidahasan@yahoo.com

**Data Availability Statement:** All relevant data are within the paper and its Supporting Information file.

## Abstract

### Introduction

Discharge against medical advice (DAMA) is an unexpected event for patients and health-care personnel. The study aimed to assess the prevalence of DAMA in neonates along with characteristics of neonates who got DAMA and, causes and predictors of DAMA.

### Methods and findings

This case-control study was carried out in Special Care Newborn Unit (SCANU) at Chittagong Medical College Hospital from July 2017 to December 2017. Clinical and demographic characteristics of neonates with DAMA were compared with that of discharged neonates. The causes of DAMA were identified by a semi-structured questionnaire. Predictors of DAMA were determined using a logistic regression model with a 95% confidence interval. A total of 6167 neonates were admitted and 1588 got DAMA. Most of the DAMA neonates were male (61.3%), term (74.7%), outborn (69.8%), delivered vaginally (65.7%), and had standard weight at admission (54.3%). A significant relationship (p < 0.001) was found between the variables of residence, place of delivery, mode of delivery, gestational age, weight at admission, and day and time of outcome with the type of discharge. False perceptions of wellbeing (28.7%), inadequate facilities for mothers (14.5%), and financial problems (14.1%) were the prevalent causes behind DAMA. Predictors of DAMA were preterm gestation (AOR 1.3, 95% CI 1.07–1.7, p = 0.013), vaginal delivery (AOR 1.56, 95% CI 1.31–1.86, p < 0.001), timing of outcome after office hours (AOR 477.15, 95% CI 236–964.6, p < 0.001), and weekends (AOR 2.55, 95% CI 2.06–3.17, p < 0.001). Neonates suffering from sepsis (AOR 1.4, 95% CI 1.1–1.7, p< 0.001), Respiratory Distress Syndrome (AOR 3.1, 95% CI 1.9–5.2, p< 0.001), prematurity without other complications (AOR 2.1, 95% CI 1.45–

**Funding:** The authors received no specific funding for this work.

**Competing interests:** The authors have declared that no competing interests exist.

3.1, p < 0.001) or who were referred from north-western districts (AOR 1.48, 95% CI 1.13–1.95, p = 0.004) had higher odds for DAMA.

## Conclusions

Identification of predictors and reasons behind DAMA may provide opportunities to improve the hospital environment and service related issues so that such vulnerable neonates can complete their treatment. We should ensure better communication with parents, provide provision for mothers' corner, especially for outborn neonates, maintain a standard ratio of neonates and healthcare providers, and adopt specific DAMA policy by the hospital authority.

## Introduction

The terms "discharge against medical advice," "leave against medical advice," and "self-discharge" all have been used to mean disregarding the doctor's advice and leaving the hospital early without caring about what might happen. Not all sick infants are fortunate enough to be admitted to the hospital [1] and when they are, it is a lost opportunity for them to leave before receiving all of their care.

The neonatal period, a child's first month of life, is the most critical time for survival [2]. Self-discharge poses a unique challenge to neonates, as they not only have emotional and cognitive immaturity but also have no legal rights to decide for themselves [1,3]. When a patient or caregiver decides to leave the hospital against the consent of the managing physician, a conflict arises between the principle of autonomy and beneficence. Which of them takes priority is the question [4,5]. Again, the signed DAMA consent form by the caregiver may help, but it does not make healthcare providers and hospitals immune from legal implications [6].

The rate of DAMA in pediatric wards reported in recent years ranged from 1.2% to 31.7% [7]. Studies have documented a higher rate of DAMA in developing than developed countries [5,8]. Discharges against medical advice (DAMA) have an impact on mortality and readmission rates in both the pediatric and adult populations [9–11]; however, they may be significantly higher in neonates. There have been a few studies done in different NICUs internationally [1,12–14], but there is a need for more data in Bangladesh. Bangladesh aims to reduce neonatal mortality to at least 12 per 1,000 live births by 2030 to meet the target of the Sustainable Development Goals of the United Nations. So quality care is needed for every neonate to reduce morbidity and mortality [15]. The study aims to ascertain the prevalence of DAMA among newborns admitted to a tertiary care hospital in Bangladesh, as well as the causes and identifying factors predicting DAMA to improve adverse outcomes of neonates.

## Materials and methods

### Study settings

This case-control study was conducted over a period of six month from July 2017 to December 2017 at the SCANU in Department of Neonatology, Chittagong Medical College Hospital. It is one of the largest SCANUs in the country, with 32 government allocated beds and 56 locally arranged beds. The average number of admitted neonates ranges from 160–210 per day in 100 cots. Admission criteria are gestational age less than 34 weeks, birth weight less than 1800 g, and any sick neonate irrespective of gestational age and birth weight. Neonates from the entire

division are referred to this SCANU, which covers a total area of 33,909.00 km2. As a tertiary and referral hospital, admission of sick neonates can not be denied in the SCANU.As a result, more than one neonate share the same cot. During study period, five consultants including two professors were working in the SCANU. There were 8–10 mid-level doctors who were on rotation from pediatrics as part of their neonatal training, and approximately 40 senior staff nurses were working in three shifts.

The Chattogram division (Fig 1), located in Bangladesh consists of 11 districts and 99 sub-districts [16]. Chattogram district serves as the administrative zone. In this study, the division was classified into six areas based on administrative zone, geographic characteristics, and access to healthcare facilities for neonates. These are Chattogram district urban (city corporation area), Chattogram district non-urban (14 sub-districts apart from city corporation area), North-western districts (6 districts), Hill tracts (3 upland districts), Cox's Bazar (1 coastal district) and Islands.

## Sampling

Non-probability consecutive sampling technique was used. We have enrolled the data of all neonates admitted in this study period on the day of their outcome. We defined DAMA cases as any neonate whose caregivers removed them from the hospital despite the attending physician recommending that they remain. Controls were those who were discharged on the advice of the attending physician. The number of neonatal admissions in this period was 6167. We excluded 1362 neonates who died in SCANU. Of the remaining 4805 neonates, a total of 3217 neonates were discharged with the medical advice of attending physicians. There were 1588 parents or guardians seeking DAMA. They all were included to avoid selection bias by the interviewers.

The neonates of case and control groups were matched on demography, geo-economic backgrounds or birth related variables. All controls included in this study had to meet the criteria of having minimum one individual with comparable age, sex, mode and place of delivery and diagnosis though the ratio of case: control was 1: 2.36. For every DAMA, at least two controls were available.

## Procedure of data collection

Data was collected through a questionnaire. A questionnaire was semi-structured including demographic profile, clinical diagnosis and reasons given by the parents for DAMA. The questionnaire was piloted on five female and five male guardians and was modified accordingly. The content of the questionnaire was developed and modified based on a literature search, a focus group discussion, and a pilot study. We mentioned all the opinions found from the previous steps. If parents chose a different reason other than mentioned options, they had to write it. At first, physicians and senior staff nurses counseled them to complete treatment, informing the prognosis and danger of discontinuation of treatment. Those intent on taking DAMA after counseling had to fill up the questionnaire where the specific cause of self-discharge had to be chosen or written by themselves or the attending senior staff nurse. The respondents usually took up to 10 minutes to complete the questionnaire.

## Ethical consideration

The research received approval from the "Ethical Review Committee" of Chittagong Medical College, Bangladesh. All participants signed an informed consent form, and confidentiality was maintained.

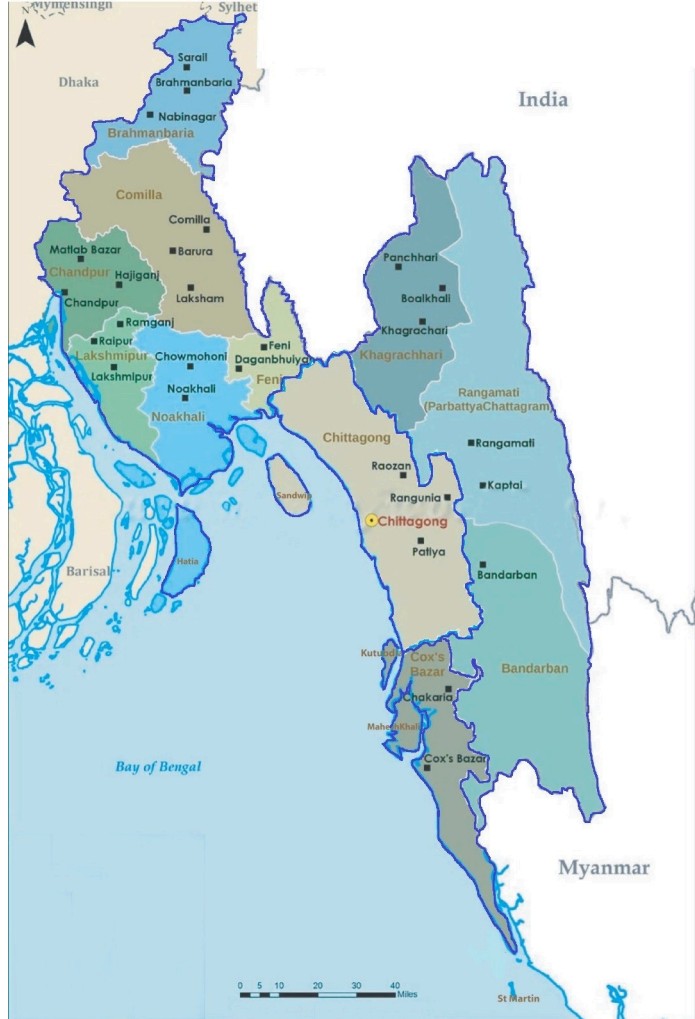

**Fig 1. Map of Chattogram division.**

## Data analysis

The dependent measure or outcome variable for the present study was the discharge of the neonates which was classified into two categories: DAMA and discharge with medical advice. Data analysis was conducted in multiple phases. In the first phase, a simple descriptive analysis (frequency and percentage) was undertaken to evaluate demographic variables such as gender, residence, mode of delivery, place of delivery, and time and day of the outcome. In the second phase, statistical analysis was performed to explore the association between discharge type and other related variables. Associations between categorical variables were assessed using a chi-square or Fisher exact test where appropriate. We compared the characteristics of the two groups using the chi-square test ($\chi 2$). We used binary logistic regression to model the correlates of discharge against medical advice. To assess the accuracy of the model, a goodness-of-fit test was performed, which provided insight into how well the model fits the data. We included each characteristic in the regression model that was significant ($p = 0.05$) in the analysis in bivariate comparison. We report adjusted odds ratio (AOR) and confidence intervals (CIs) from this model.

## Results

A total of 6167 neonates were admitted during six month study period from July 2017 to December 2017. Bed occupancy rate was 141.2% in the study period. Guardians of 1588 neonates discharged the neonate against medical advice even after being counseled by attending physicians and nurses. Among these neonates discharged against medical advice, 63.5% were admitted on 1st day of their life. SCANU is the tertiary care and referral center for neonates of greater Chattogram. Only 30.4% (n = 482) of DAMA neonates were referred from urban areas. Forty-eight percent (48.4%, n = 769) of neonates were admitted from outside the city corporation area. Two hundred nine patients (13.2%) were from north-western districts. Referred neonates from hill districts (3.1%), Cox's Bazar (3.2%) and different islands (1.8%) were also self-discharged. The maximum numbers (61.3%, n = 974) of neonates were male. The male-to-female ratio was 1.6: 1. About 69.8% (n = 1108) of neonates were outborn. Most neonates (54.3%, n = 863) had 2500 g or more weight at admission. Those delivered at term were 1186 (74.7%), while 402 (25.3%) were preterm. Spontaneous vaginal delivery comprised 65.7% (n = 1044). Out of the total DAMA neonates, 42% (n = 667) left the ward in the evening and 10% (n = 161) at night. The percentage of DAMA was less (31.1%) on weekdays among total discharges, whereas it was more (47.2%) on weekly holidays and festive times. The mean hospital stay of DAMA patients was 3.66±3.95 days. One-third of DAMA patients (32.8% n = 521) were self-discharged within the first day of admission, 32.4% (n = 514) during 2–3rd hospital days, 24.2% (n = 384) during 4–7 days and 10.6% (n = 169) had eight days or more hospital stay (Table 1).

Common co-morbidities of DAMA neonates were perinatal asphyxia (41.9%, n = 664), sepsis (33.7%, n = 536) and prematurity without any other complications (10.6%, n = 169). Other diagnoses were neonatal jaundice, respiratory distress syndrome (RDS), infant of diabetic mother (IDM), transient tachypnea of newborn (TTN) and necrotizing enterocolitis (NEC) (Table 2). The percentage of congenital anomaly among DAMA neonates was 4.4% (n = 71), mostly in the form of congenital heart disease (1.3%), cleft palate±cleft lip (0.6%), Down syndrome (0.4%) and multiple congenital anomalies (0.4%).

The percentage of DAMA was calculated at 25.7% in this work. Patient-related causes (40.2%, n = 639) were prevalent behind DAMA. Hospital-related (33.4%, n = 529) and family-related causes (23.1%, n = 367) were also noted. Common reasons identified for DAMA were the false sense of parents about the wellbeing of their babies (28.7%, n = 455), lack of facilities for the mother (14.5%, n = 231), financial problems (14.1%, n = 233), dissatisfaction with treatment (10.9%, n = 173), baby can take oral feeding (7.4%, n = 117), lack of attendants (6.9%, n = 110), fewer facilities for attendants (6.9%, n = 69). Guardians had a fear of observing the death of other sick neonates in SCANU (3.5%, n = 56) and a lack of hope for further improvement (4.2%, n = 67) (Table 3). Parents who self-discharged their babies due to a false perception of wellbeing mostly suffered from asphyxia (45.7%) and sepsis (33.4%). There was a significant difference in family-related (p < 0.001) and patient-related (p <0.001) causes of DAMA between preterm and term neonates. Only thirteen caregivers had multiple causes to seek DAMA. Among these multiple respondents, two caregivers had financial problem and they also lost their hope for clinical improvement in the neonate. Six caregivers said about the false sense of well-being with the lack of facilities for mothers/ attendants. Five caregivers were unsatisfied with the treatment and also mentioned the lack of facilities for mothers.

The chi-square test was used to study the relationship between variables with the type of discharge. A significant relationship was observed between the variables of residence (p <0.001), place of delivery (p <0.001), mode of delivery (p <0.001), gestational age (p <0.001), weight at admission (p <0.001), day and time of outcome (p <0.001) with the type of

**Table 1. Demographic profile of neonates DAMA and discharge with medical advice.**

| Variable | | DAMA<br>n = 1588 | Discharge<br>n = 3217 | P value |
|---|---|---|---|---|
| Age at admission | Day 1 | 1009 (63.5%) | 2062(64.1%) | 0.61 |
| | Day 2–7 | 353 (22.2%) | 704(21.9%) | |
| | Day 8–15 | 87 (5.5%) | 176(5.5%) | |
| | Day 16–21 | 70 (4.4%) | 161(5%) | |
| | Day 22–28 | 69 (4.4%) | 114(3.5%) | |
| Residence | Chattogram Urban[1] | 482(30.4%) | 905(28.1%) | <0.001 |
| | Chattogram Non urban[2] | 769(48.4%) | 1748(54.3%) | |
| | Hill Tracts[3] | 49(3.1%) | 85(2.6%) | |
| | Cox's Bazar | 51(3.2%) | 117(3.6%) | |
| | North-western Districts[4] | 209(13.2%) | 281(8.7%) | |
| | Islands[5] | 28(1.8%) | 81(2.5%) | |
| Gender | Male | 974 (61.3%) | 2050 (63.7%) | 0.107 |
| | Female | 614 (38.7%) | 1167 (36.3%) | |
| Place of Delivery | Inborn | 480 (30.2%) | 1200 (37.3%) | <0.001 |
| | Outborn | 1108 (69.8%) | 2016 (62.7%) | |
| Mode of delivery | Pervaginal | 1044 (65.7%) | 1805 (56.1%) | <0.001 |
| | LSCS | 544 (34.3%) | 1411 (43.9%) | |
| Gestational age | Preterm | 402 (25.3%) | 470 (14.6%) | <0.001 |
| | Term | 1186 (74.7%) | 2747 (85.4%) | |
| Birth Weight | <2500 g | 725(45.7%) | 1188 (36.9%) | <0.001 |
| | ≥2500 g | 863 (54.3%) | 2029 (63.1%) | |
| Day of outcome | Working Day | 1314 (82.7%) | 2910 (90.5%) | <0.001 |
| | Holiday/Festive | 274(17.3%) | 306 (9.5%) | |
| Time of outcome | Morning | 760(47.9%) | 3209 (80.9%) | <0.001 |
| | Evening | 667(42%) | 08(0.2%) | |
| | Night | 161(10.1%) | 00(0%) | |
| Duration of hospital stay | ≤1 day | 521(32.80%) | 655(20.4%) | <0.001 |
| | 2–7 days | 898(56.5%) | 1828(56.8%) | |
| | 8–14 days | 127(8%) | 546(17%) | |
| | 15–21 days | 29(1.8%) | 131(4.1%) | |
| | 22–28 days | 07 (0.4%) | 51 (1.6%) | |
| | >28 days | 06(0.4%) | 06 (0.2%) | |

1 = Chattogram Urban (City corporation area).

2 = Chattogram Non urban (Outside city corporation area of Chattogram, 14 upazilas).

3 = Hill Tracts (Rangamati,Khagrachari and Bandarban).

4 = North-western Districts (Brahmanbaria, Cumilla, Chandpur, Laskhmipur Noakhali and Feni).

5 = Islands (Sandwip, Hatia, Kutubdia,Moheshkhali, St,Martin).

discharge. There was no significant relationship between gender (p = 0.107) and age at admission (p = 0.61) with the type of discharge (Table 1).

The results of logistic regression to investigate the impact of independent variables on the probability of DAMA were presented in Table 4. Our logistic regression model was significantly better as it explained more of the variance in the outcome, which was reflected by the highly significant chi-square value (chi-square = 2365.3, df = 17, p < .000). Additionally, the Hosmer-Lemeshow goodness-of-fit test suggested that our model was a good fit to the data (p = 0.6,>0.05). The new model performed better than the null model, with a correct

**Table 2. Disease profile of DAMA.**

| Diagnosis | DAMA (n = 1588) | Discharge with advice (n = 3217) | Result (p value) |
|---|---|---|---|
| Perinatal Asphyxia (PNA) | 664 (41.9%) | 1596(49.6%) | <0.001 |
| Sepsis | 536 (33.7%) | 905(28.1%) | <0.001 |
| Prematurity | 169 (10.6%) | 140(4.4%) | <0.001 |
| Jaundice | 45 (2.8%) | 203 (6.3%) | <0.001 |
| Infant of Diabetes Mellitus (IDM) | 29 (1.8%) | 101 (3.1%) | 0.008 |
| Respiratory Distress Syndrome (RDS) | 79 (5.0%) | 47 (1.5%) | <0.001 |
| Transient Tachypnea of Newborn (TTN) | 17 (1.1%) | 73 (2.3%) | 0.004 |
| Necrotizing Enterocolitis (NEC) | 06 (0.4%) | 05 (0.2%) | 0.129 |
| Others | 43 (2.7%) | 147 (4.6%) | 0.002 |

classification rate of 84.3% compared to 67%. In accordance with the results presented, birth weight and place of delivery were not predictive of DAMA. Gestational age, mode of delivery and day and time of outcome influenced parents' decision to DAMA. Preterm gestation (AOR 1.3, 95% CI 1.07–1.7, p = 0.013), vaginal delivery (AOR 1.56, 95% CI 1.31–1.86, p <0.001), the timing of outcome after office hours (AOR 477.15, 95% CI 236–964.6, p <0.001) and week-ends (AOR 2.55, 95% CI 2.06–3.17, p <0.001) had greater odds of DAMA. Since the most common reason for referral was PNA, other diagnoses were assessed against PNA. Neonates with sepsis (AOR 1.4, 95% CI 1.1–1.7, p<0.001), RDS (AOR 3.1, 95% CI 1.9–5.2, p <0.001), prematurity without other complications (AOR 2.1, 95% CI 1.45–3.1, p <0.001) were found as predictors of DAMA. In terms of residence, those from north-western districts (AOR 1.48, 95% CI 1.13–1.95, p 0.001) and hill tracts (AOR 1.07, 95% CI 0.066–1.72, p 0.80) had higher odds for DAMA. Yet, only north-western districts were statistically significant as a residence compared to the Chattogram city corporation area (Table 4).

**Table 3. Cause behind DAMA according to care-givers' response.**

| | Individual Cause | Preterm (n = 402) | Term (n = 1186) | Total (n = 1588) | P value |
|---|---|---|---|---|---|
| Patient-Related (n = 639) 40.2% | False sense of wellbeing | 85(21.1%) | 370(31.2%) | 455(28.7%) | <0.001 |
| | Baby can take oral feeding | 17 (4.2%) | 100 (8.4%) | 117(7.4%) | |
| | No improvement, poor prognosis | 19 (4.7%) | 48 (4%) | 67 (4.2%) | |
| Hospital-Related (n = 529) 33.4% | Lack of Facilities for mother | 68 (16.9%) | 163 (13.7%) | 231(14.5%) | 0.706 |
| | Lack of facilities for attendants | 19 (4.7%) | 50 (4.2%) | 69(6.9%) | |
| | Not satisfied with treatment | 37 (9.2%) | 136 (11.5%) | 173(10.9%) | |
| | Fear of observing death of other sick neonates | 13 (3.2%) | 43 (3.6%) | 56(3.5%) | |
| Family-Related (n = 367) 23.1% | Financial Problem | 92 (22.9%) | 131(11%) | 223 (14.1%) | <0.001 |
| | Lack of attendants | 36(9%) | 74 (6.2%) | 110 (6.9%) | |
| | Mother seriously ill | 03 (0.7%) | 12 (1%) | 15(0.9%) | |
| | Death of mother | 05 (1.2%) | 06 (0.5%) | 11(0.7%) | |
| | Unable to shift the neonate to referral center | 0 (0%) | 08 (0.7%) | 08(0.5%) | |
| Others (n = 53) 3.3% | | 8(2%) | 45(3.8%) | 53(3.3%) | 0.08 |

**Table 4. Logistic regression to analyze the influence of independent variables.**

|  |  | Adjusted Odds Ratio (AOR) | 95% CI | P value |
|---|---|---|---|---|
| Gestational age | Preterm | 1.37 | 1.07–1.76 | 0.013 |
| Mode of Delivery | Vaginal Delivery | 1.56 | 1.31–1.86 | 0.00 |
| Day of Outcome | Holiday-Festive | 2.55 | 2.06–3.17 | 0.00 |
| Time of Outcome | Evening-Night | 477.15 | 236.0–964.6 | 0.00 |
| Diagnosis | Sepsis | 1.421 | 1.175–1.720 | 0.00 |
|  | Jaundice | 0.683 | 0.442–1.054 | 0.085 |
|  | Preterm without complication | 2.104 | 1.446–3.063 | 0.000 |
|  | IDM | 0.742 | 0.411–1.339 | 0.322 |
|  | RDS | 3.163 | 1.914–5.224 | 0.000 |
|  | TTN | 1.001 | 0.534–1.878 | 0.998 |
| Residence | Chattogram (Non Urban) | 0.792 | 0.656–0.956 | 0.015 |
|  | Hill tracts | 1.064 | 0.657–1.722 | 0.802 |
|  | Cox's Bazar | 0.551 | 0.326–0.933 | 0.027 |
|  | North-western districts | 1.488 | 1.132–1.955 | 0.004 |
|  | Islands | 0.683 | 0.377–1.236 | 0.208 |

## Discussion

DAMA is a major public health issue, poses an immediate risk to the life of the neonate [3]. The prevalence of DAMA in this study was found to be very high at 25.7%. This finding is similar to the 22.24% and 25.4% reported from NICUs in India [12,13] but higher than Nepal (18%) [1] and Nigeria (11.2%) [3]. Emergency admission and admission to the neonatal intensive care unit itself was reported as a significant independent risk factor for DAMA [2,8]. Prejudiced decision making by the family members contributed to increase DAMA among neonates [17]. The rate of DAMA differs with study setting; the time of the study, economic status and socio-cultural factors also influence the rate [13,18]. A retrospective review of 10 years of medical records of neonates found only 1.6% DAMA at a university hospital NICU in Saudi Arabia [18].

There was no significant gender bias among DAMA cases. This is consistent with the previous literature [12,19,20]. About two-thirds of the DAMA infants (65.7%) were born vaginally, which made vaginal delivery a potential risk for DAMA. This finding matched with previous studies in Saudi Arabia [18]. and western Nigeria [3] The higher DAMA was partly due to the early ambulation and discharge of the mothers compared to cesarean section. Bosco et al found that mother's antenatal check-up and parity influenced DAMA [10]; although we didn't collect such data. Unlike other NICUs at home and abroad, most of the DAMA neonates were outborn in our study. It was challenging for the neonates to stay here because their moms were either at another hospital or home. Most of the DAMA neonates had full-term gestation and standard weight at admission which matched with studies in India and Saudi Arabia [13,18]. Many parents had the wrong perception that a baby who was big and born at term was healthy [18].

DAMA was more on festive and weekends than any other days of the week. Similar findings were found by Kumar et al. and Turkistani et al. [13,18] Discharge decisions were typically made by the mid level doctors with the consent of consultants on the morning following the round. Due to a lack of staff and hospital policy, the discharge rate with advice was lower in the evening and almost zero at night. It could explain the increase in DAMA after office hours.

Most of the neonates admitted in SCANU at their 1st day of life. Percentage of admission gradually decreased with the age of the neonates. There was no significant variation between

discharge and DAMA group in this regard. Our study found higher rate of DAMA (90%) in first week; importantly, one-third of DAMA happened within the first 24 hours. Similarly 92.5% DAMA happened at a NICU in Lahore with 29.9% within the first 24 hours [17]. Newborns are at greater risk for illness in the first few days of life. The seriousness is augmented when dealing with a newborn at risk who is taken away from the intensive care unit against medical advice. On the other hand, only 20.4% discharge decisions were made within first 24 hours. The rate of discharge in first week was 77.2%. The percentage of DAMA in first week was 56%-73% in previous study [1,3,11,17,18]. The rituals of naming ceremony for neonates, typically held at seven days of age, may have contributed to the increased rate of DAMA during the first week [1,3].

In our study, we classified the residences of neonates into six classes and distinguished between urban and non-urban areas only within the Chattogram district. Around half (48.4%) of the DAMA neonates were admitted from non-urban areas, and 21.3% were from outside the Chattogram district, which contributed two-thirds of the DAMA. The residence of neonates reflects the availability of neonatal treatment facilities in the region. Apart from this particular SCANU, the other government SCANUs available in this division were in Cox's Bazar and Bandarban. It is worth noting that there were also private SCANUs available in the area, although they may have limited capacity and higher costs compared to government facilities. The number of DAMA patients was higher from rural areas in India [19].

DAMA is a major challenge for hospital treatment teams and it is practically common in all medical facilities and difficult to eliminate leading to unpredictable complications [2]. PNA, sepsis and prematurity without complications were the most common diagnoses in DAMA neonates. These diagnoses have also been identified by World Health Organization as the greatest cause of mortality of newborns in the developing world. Sepsis, PNA, respiratory distress, and congenital heart disease were the most prevalent diagnostic related categories in a study of India [12]. Prematurity (25.8%) exceeds PNA (23.9%) and sepsis (22.6%) in Nigeria [3]. Another two studies found additional causes of low birth weight (10.2%) and neonatal jaundice (10.2%) [13]: major congenital malformations (58%) and perinatal asphyxia (14%) [21]. Common congenital anomalies which lead to DAMA were congenital anorectal malformations, esophageal atresia/tracheoesophageal fistula, and congenital intestinal atresia [2].

In many studies, non-affordability or financial constraints were the most common reason for taking self-discharge [1,10,12,19]. Like a study done in Iran [20], we found false perceptions of the wellbeing of neonates by the parents as the most typical cause of DAMA and it was primarily found in term neonates. One-third of such neonates were treated and improved with supportive care and antibiotics for sepsis, so parents had wrong idea of taking discharge early. We had a few numbers of midlevel doctors and senior staff nurses. The average bed occupancy rate in the study place was above 100 percent. As all sick neonates were allowed for admission, we were overburdened with a huge number of neonates. The communication between health care providers and parents may not be the standard. Fortunately, the percentage of dissatisfaction (10.9%) was not much. Lack of facilities for mothers and attendants contributed 20% to DAMA, which was quite reasonable. Mothers require comfort after giving birth to a baby [20]. The hospital authority couldn't provide separate spaces for the mother and attendants. Most of the attendants and some mother had to sit and recline on the ground outside the SCANU after visiting hours. Financial constraints were the major cause of DAMA among preterm neonates. It may be explained by the fact that they required intensive care and a more extended hospital stay than term neonates. Belief in the incurability of the diseases, multiple malformations, concern about the prognosis of the diseases, family pressure, sociocultural thoughts, and inappropriate behavior with patients by hospital team were the causes of DAMA found in different studies [2,12,17,18,22]. False perception of wellbeing (34.4%),

lack of facilities for mothers (15%) and dissatisfaction (13.6%) were the major causes allowed them to take rapid decisions about DAMA on the 1st day.

In our study, neonates who delivered vaginally had 1.56 times more chance for DAMA than neonates delivered by cesarean section. Though there was an increased percentage of out-born and term neonates having standard weight at admission among self-discharged neonates, it was not found to be predictors of DAMA in regression analysis. Rather, preterm neonates had higher odds (AOR = 1.37 95% CI 1.07–1.7, p = 0.013) for DAMA than term neonates. In our analysis, prematurity was a variable of gestation but was not noted as distinct diagnostic category as most of the preterm neonates were admitted with various complications such as asphyxia, sepsis, respiratory distress and they were entitled accordingly. Interestingly, when they were admitted only due to prematurity without any complications, the risk of DAMA increased (AOR 2.1, 95% CI 1.45–3.1, p <0.001) Neonatal sepsis (AOR 1.4, 95% CI 1.1–1.7, p<0.001) and Respiratory distress syndrome (AOR 3.1, 95% CI 1.9–5.2, p <0.001) were found as other predictors of DAMA in our study. Neonates often require longer stays in the NICU to complete antibiotic protocols, control seizures, and establish feeding. However, caregivers may have difficulty accepting these extended stays, especially when compared to the shorter hospital stays commonly seen in adults. Moreover, caregivers were not interested to complete the antibiotic regimen when neonates showed transient improvement after starting initial management. The grave outcomes of neonates may persist when cases of culture-proven sepsis go undetected due to DAMA. Bosco et al. found low birth weight (AOR 2.1, 95% CI 1.7–9.4, P = 0.030), low APGAR score (AOR 6.9, 95% CI 6.9, 95% CI 2.1–15.7, P < 0.001) and non life threatening congenital malformation (AOR 2.1, 95% CI 1.12.3, P = 0.005) as predictor of DAMA [10].

Turkistani et al found that the prevalence of DAMA was influenced by time factors (weekend and season). Wednesdays (27.3%) and the month of May (33%) individually represented one-third of all DAMA infants [18]. We also found substantial impact of holidays (festive and weekends) and the timing after office hour on DAMA. We didn't obtain month wise variation due to short study period. Neonates admitted from north-western districts had higher odds for DAMA. These six districts were situated at a minimum distance of 92 kilometers away from the Chattogram district. Since there was no government SCANU in these districts, neonates were managed in private hospitals. When the situation worsened or due to family demands, neonates were usually referred, and often without their mothers. Previous studies have addressed the role of residence in other districts and long distance between hospital and home in cases of DAMA [18,20].

## Limitation

Although we included all neonates within a specific time range, this was a single center study. After DAMA, we were unable to monitor the newborns, and we lacked precise information on their prognosis. No comprehensive maternal information was recorded, which could be a contributing factor to DAMA.

## Conclusion

DAMA is a complex issue and needs attention. This study helped us learn more about DAMA in-depth. The prevalence of DAMA in this work is not less. False perception of the wellbeing of neonates, fewer facilities for mothers and financial problems are the leading causes behind DAMA in this study. Prematurity, vaginal delivery, timing of DAMA, residence at distant place was found to be important predictors. Neonates suffering from sepsis and RDS had more chance of DAMA. So care should be taken to handle such neonates including counseling

of parents regarding completion of treatment on admission. A standard ratio of neonates and healthcare providers should be maintained for better management of neonates and better communication with the parents. We propose to arrange a separate corner for maternal rest and accommodation which may reduce the rate of DAMA. Hospitals should develop specific DAMA policies to safeguard healthcare professionals.

## Supporting information

**S1 File. Data July to December.**
(SAV)

## Acknowledgments

The authors would like to thank all the physicians and senior staff nurses placed in SCANU, Chittagong Medical College Hospital, who helped gather data and provide guidance to the parents of neonates who were admitted.

## Author Contributions

**Conceptualization:** Syeda Humaida Hasan, Jagadish Chandra Das.

**Data curation:** Syeda Humaida Hasan, Jagadish Chandra Das, Kamrun Nahar, Muhammad Jabed Bin Amin Chowdhury, Tamanna Zahur, Mohammad Abu Faisal, Zabeen Choudhury, Dhiman Chowdhury.

**Formal analysis:** Syeda Humaida Hasan, Tamanna Zahur.

**Methodology:** Jagadish Chandra Das, Kamrun Nahar, Muhammad Jabed Bin Amin Chowdhury, Tamanna Zahur, Mohammad Abu Faisal, Zabeen Choudhury, Dhiman Chowdhury.

**Project administration:** Syeda Humaida Hasan.

**Supervision:** Syeda Humaida Hasan, Jagadish Chandra Das.

**Writing – original draft:** Syeda Humaida Hasan, Kamrun Nahar, Tamanna Zahur, Mohammad Abu Faisal, Zabeen Choudhury.

**Writing – review & editing:** Syeda Humaida Hasan, Jagadish Chandra Das, Muhammad Jabed Bin Amin Chowdhury, Tamanna Zahur, Dhiman Chowdhury.

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
