## [Decision Letter · Decision Letter 0]

7 Feb 2023

PONE-D-23-01392Discharge against medical advice in Special care Newborn Unit in Chattogram, Bangladesh: Prevalence, causes and predictorsPLOS ONE

Dear Dr. Hasan,

Thank you for submitting your manuscript to PLOS ONE. After careful consideration, we feel that it has merit but does not fully meet PLOS ONE’s publication criteria as it currently stands. Therefore, we invite you to submit a revised version of the manuscript that addresses the points raised during the review process.

ACADEMIC EDITOR: 

As stated by the reviewers, there are important issues that need to be addressed:

The manuscript requires English language copy editThe tables need to be revised as stated by reviewer 2The discussion needs to show the implication of the findings 

We look forward to receiving your revised manuscript.

Kind regards,

Atnafu Mekonnen Tekleab, M.D

Academic Editor

PLOS ONE

Journal Requirements:

Reviewers' comments:

Reviewer's Responses to Questions

**Comments to the Author**

1. Is the manuscript technically sound, and do the data support the conclusions?

Reviewer #1: Yes

Reviewer #2: Partly

Reviewer #3: Yes

2. Has the statistical analysis been performed appropriately and rigorously? 

Reviewer #1: Yes

Reviewer #2: No

Reviewer #3: Yes

3. Have the authors made all data underlying the findings in their manuscript fully available?

Reviewer #1: Yes

Reviewer #2: Yes

Reviewer #3: Yes

4. Is the manuscript presented in an intelligible fashion and written in standard English?

Reviewer #1: Yes

Reviewer #2: Yes

Reviewer #3: No

5. Review Comments to the Author

Reviewer #1: Authors have done a fair job in presenting the study findings. Few clarifications are needed.

1. Data seems to be more than 5 years old which make the findings less relevant.

2. Sample size has not been calculated

3. Table 2 and 3 should mention p value and statistics for each variable.

4. The goodness of fit should be mentioned in statistics paragraph

5. All conclusions should be based on the study results only. Terminology like medico legal etc should be supported by evidence from the study results.

6. There maybe many other reasons for dama or multiple causes in the same patient which should be reported and accounted for

7. Discussion should be re organised to explain the results logically.

8. Long term outcomes of DAMA could have been reported.

9. Report should follow Strobe checklist

10. A case control methodolgy would have been more sound in design of the study

Reviewer #2: The authors report a cross-sectional study of neonates who were discharged against medical advice at a single site in Bangladesh. They described the characteristics of neonates who were DAMA and compared those to those neonates who did not DAMA. This is an interesting area of investigation but I believe it could be greatly strengthened. My major comments are as follows:

-There needs to be more of a rationale included on the content of the questionnaire administered to caregivers.

-Similarly, the rationale for variables assessed to compare neonates who DAMA to those who did not is needed.

-The data only cover a six-month period that is nearly 5 years ago at this point. The rational for this time selection should be clearly explained.

-I think including a multivariable regression model would greatly strengthen their case.

-There is no limitations paragraph. No study is perfect but the limitations paragraph gives the authors an opportunity to acknowledge the limitations of the current study as well as call for additional work to address the limitations. I strongly suggest including one.

-Finally, I think this article should be carefully read and reviewed by a native English speaker to ensure the syntax and grammar are correct.

Additional comments and recommendations by section are provided below.

Abstract:

-Minor point, but it should be clear that the aim is to assess characteristics of neonates who DAMA and the causes and predictors of DAMA.

-The line “A total of 6167 neonates were admitted and 1588 were self-discharged.” needs some clarification. How many were discharged against medical advice? Is that the same as “self-discharge”? I suggest being consistent with the language.

-The Methods of the Abstract should include how causes were identified.

-“Variable of residence” is vague. Does this mean urban vs rural? Needs to be clarified.

-Something for the authors to consider: is a neonate who was the product of a term, vaginal delivery with standard birthweight neonate really a manifestation of a “false perception” as the authors present? I ask because those are all low-risk characteristics for untoward events in the neonatal period.

-The odds ratio for “timing of outcome after office hours” is gigantic. Is this true? I don’t think I have ever seen an odds ratio of that magnitude.

-The sentence “Diagnosis and residence also played roles in their decision making” is vague and needs to be reworded. Which direction was the relationship?

Introduction:

-The sentence “So, treating doctors still worry about getting into trouble with the law.” is unclear. Are doctors worried about being sued when treating neonates any more so than treating other patient populations? If so, why?

-Again, why use “self-discharge” and “DAMA” throughout the paper? It is best to be consistent.

Methods:

-More of a description of the study setting would be helpful to the reader to see how these results do, or do not, compare to settings where they may work.

-Similarly, a description of the indications for SCANU admission is needed. By the description, it seems that many of the neonates were term, normal birthweight, etc., which makes me wonder why they would need much care at all.

-In order to demonstrate rigor, the content of the questionnaire administered to caregivers needs some basis of the content. I applaud the authors for piloting it with 10 caregivers, but where did the content come from? Ideally, it should be a validated tool, but in the absence of such, pointing at the results of prior studies may suffice.

-Also, was this survey quantitative or qualitative? It appears to be quantitative but again, the rationale for the content and how the questions were posed is needed.

-Was there any reason a multivariable logistic regression approach was not taken? Yes, bivariate comparisons are nice, but it is stronger to show which factors are independently associated with the outcome adjusting for potential confounders whenever possible.

Results:

-Referring to places like “Cumilla” and “Cox’s bazar” is not helpful for the reader as it implies they know where these are. I suggest describing these settings instead. Are they urban, rural, etc.?

-The comparison made by length of stay needs for DAMA vs not should be by median length of stay, not chi square. I suggest this because the description of “term, vaginal delivery with standard birthweight neonate” among those who DAMA is descriptive of a healthy population. So it makes me wonder if they needed to be in the hospital longer than a day, or why such a neonate is admitted to the SCANU.

-The “Result (Chi square)” probably isn’t necessary to include. I don’t think I have seen this in any paper. The p value for a Chi square test is sufficient.

-Table 2: Please spell out all abbreviated diagnoses for clarity.

-Table 2: I defer to the editor and a biostatistical reviewer, but a single Chi square for this entire table is misleading. I suggest comparisons be made by each row instead.

-Why is the “percentage of DAMA estimated as 25.7%”? That seems like a precise proportion and not an estimate.

-Table 3: Please clarify in the title that these were ascertained from caregivers.

-Table 3: Again, I think row by row comparisons are more meaningful than one for the entire table.

-Table 4 does not need the beta coefficients or standard errors. Odds ratios, 95% CIs +/- P values should suffice.

Discussion:

-Is DAMA a “cause of patient dissatisfaction” or a manifestation of it? I suspect it is the latter.

-The first part of paragraph 2 would be better suited in a description of the Study Setting in a section in the Methods.

-In general, the Discussion reads as a bit rambling and could be focused. I suggest the authors spend time thinking more about the implications of each finding and trying to tie that into each paragraph.

-Is the final paragraph a Conclusions paragraph? If so, I suggest clearly labeling it as such.

Reviewer #3: Main finding of the paper

Study period-6 months (from July-December 2017)

• Bed occupancy rate -141 %( unacceptably high)

• One thousand five hundred eighty (25.7%) of the patients discharge against medical advice (DAMA) among 6167 admitted neonate in the study period the prevalence is very high.

• Main reason for DAMA were false perception of well-being of neonates, inadequate facility for mothers and financial reason

• Main predictors for DAMA were preterm delivery, vaginal delivery, and timing of outcome after office hours

No major gaps identified in abstract, introduction and methods section)

Result section:

• The authors used old data set (2017)

• The description of the result is not coherent and the sentences are fragmented it is not eye caching(need rewriting)

• Better to put the main finding of tables in summarized ways (as long as you annexed the table within the paragraph)

• While summarizing the result, mentioning percentage (%) alone may not give sense. so it is better to report as Number (%) uniformly.

6. PLOS authors have the option to publish the peer review history of their article (what does this mean?). If published, this will include your full peer review and any attached files.

Reviewer #1: **Yes: **Dr Shashidhar A

Reviewer #2: No

Reviewer #3: No

---

## [Author Response · Author response to Decision Letter 0]

22 Mar 2023

Reviewer 1: I have incorporated all of your suggestions into my revision.They were very helpful. Thank you.

Reviewer 2: I have incorporated all of your suggestions into my revision. Thank you for your help. 

Reviewer 3: I have incorporated all of your suggestions into my revision.They were very helpful. Thank you.

---

## [Editor Report · Decision Letter 1]

23 Mar 2023

PONE-D-23-01392R1Discharge against medical advice in Special Care Newborn Unit in Chattogram, Bangladesh: prevalence, causes and predictorsPLOS ONE

Dear Dr. Hasan,

Thank you for submitting your manuscript to PLOS ONE. After careful consideration, we feel that it has merit but does not fully meet PLOS ONE’s publication criteria as it currently stands. Therefore, we invite you to submit a revised version of the manuscript that addresses the points raised during the review process.

ACADEMIC EDITOR:- During your initial submission of your manuscript, you mentioned that the study design was cross-sectional study design. However, in the revised manuscript, you have stated that "case-control" study design was used. How the change in the study design was made during manuscript write up stage is not clear. Please clarify it.- Please remove map of Bangladesh from the manuscript. It doesn't have direct relationship with your research findings. - Your description of the study setting has to be concise.- In your findings of Binary Logistic Regression, please use Adjusted Odds Ratio (AOR) not  the crude Odds Ratio.

We look forward to receiving your revised manuscript.

Kind regards,

Atnafu Mekonnen Tekleab, M.D

Academic Editor

PLOS ONE
---

## [Author Response · Author response to Decision Letter 1]

4 Apr 2023

I have incorporated all of your suggestions into my revision. Thank you for your help.

---

## [Editor Report · Decision Letter 2]

6 Apr 2023

Discharge against medical advice in Special Care Newborn Unit in Chattogram, Bangladesh: prevalence, causes and predictors

PONE-D-23-01392R2

Dear Dr. Hasan,

We’re pleased to inform you that your manuscript has been judged scientifically suitable for publication and will be formally accepted for publication once it meets all outstanding technical requirements.

Kind regards,

Atnafu Mekonnen Tekleab, M.D

Academic Editor

PLOS ONE
---

## [Editor Report · Acceptance letter]

11 Apr 2023

PONE-D-23-01392R2 

Discharge against medical advice in Special Care Newborn Unit in Chattogram, Bangladesh: Prevalence, causes and predictors 

Dear Dr. Hasan:

I'm pleased to inform you that your manuscript has been deemed suitable for publication in PLOS ONE. Congratulations! Your manuscript is now with our production department. 

Kind regards, 

on behalf of

Dr. Atnafu Mekonnen Tekleab 

Academic Editor

PLOS ONE